

# High nonpublication rate from publication professionals hinders evidence-based publication practices

Luke C. Carey[1], Serina Stretton[1], Charlotte A. Kenreigh[2], Linda T. Wagner[2] and Karen L. Woolley[1,3,4]

[1] ProScribe—Envision Pharma Group, Sydney, New South Wales, Australia
[2] Excel—Envision Pharma Group, Southport, CT, United States
[3] University of Queensland, Brisbane, Queensland, Australia
[4] University of the Sunshine Coast, Maroochydore DC, Queensland, Australia

Corresponding author
Luke C. Carey,
luke.carey@envisionpharmagroup.com

## ABSTRACT

**Background.** The need for timely, ethical, and high-quality reporting of clinical trial results has seen a rise in demand for publication professionals. These publication experts, who are not ghostwriters, work with leading medical researchers and funders around the world to plan and prepare thousands of publications each year. Despite the involvement of publication professionals in an increasing number of peer-reviewed publications, especially those that affect patient care, there is limited evidence-based guidance in the peer-reviewed literature on their publication practices. Similar to the push for editors and the peer-review community to conduct and publish research on publication ethics and the peer-review process, the International Society for Medical Publication Professionals (ISMPP) has encouraged members to conduct and publish research on publication planning and practices. Our primary objective was to investigate the publication rate of research presented at ISMPP Annual Meetings.

**Methods.** ISMPP Annual Meeting abstract lists (April 2009–April 2014) were searched in November 2014 and data were extracted into a pilot-tested spreadsheet. MEDLINE was searched in December 2014 to determine the publication rate (calculated as the % of presented abstracts published as full papers in peer-reviewed journals). Data were analyzed using the Cochran-Armitage trend test (significance: $P < .05$) by an independent academic statistician.

**Results.** From 2009 to 2014, there were 220 abstracts submitted, 185 accepted, and 164 presented. There were four corresponding publications (publication rate 2.4%). Over time, ISMPP's abstract acceptance rate (overall: 84.1%) did not change, but the number of abstracts presented increased significantly ($P = .02$). Most abstracts were presented as posters (81.1%) and most research was observational (72.6%). Most researchers came from the US (78.0%), followed by Europe (17.7%), and the Asia-Pacific region (11.2%).

**Discussion.** Research presented at ISMPP Annual Meetings has rarely been published in peer-reviewed journals. The high rate of nonpublication by publication professionals has now been quantified and is of concern. Publication professionals should do more to contribute to evidence-based publication practices, including, and especially, their own. Unless the barriers to publication are identified and addressed, the practices of publication professionals, which affect thousands of peer-reviewed publications each year, will remain hidden and unproven.

## INTRODUCTION

Rennie and Flanagin warned that the quest to improve publication practices requires "…a massive and prolonged effort on the part of researchers, funders, institutions, and journal editors…" (*Rennie & Flanagin, 2014*). Fundamental to this quest is research on publication practices. Such research should address important questions, be well-designed, conducted, and published—in full; published abstracts are insufficient to inform practice (*Hopewell et al., 2008*). Conducting and publishing research on publication practices, however, isn't easy, even for editors and the peer-review community (*Rennie & Flanagin, 2014*). Malički and colleagues (*2014*) reported that "…39% of research presented at Peer Review and Biomedical Publication congresses (PRCs) had not been fully published…" based on an analysis of the seven congresses from 1989 to 2013.

Publication professionals work with researchers and funders around the world to plan and prepare thousands of publications each year (*Wager et al., 2014*) and have a responsibility to join the research effort. These experts, who are not ghostwriters, must shine an empirical light on the integrity and effectiveness of their practices as these practices affect the quality and currency of the medical literature that influences patient care. Unless publication professionals publish their research results in peer-reviewed journals, much of what they do remains hidden.

The International Society for Medical Publication Professionals (ISMPP), a not-for-profit association with >1,400 members, has a mission to "advance the medical publication profession globally through enhanced integrity and transparency in medical publications, improved standards and best practices, and education, advocacy, and professional collaborations" (http://www.ismpp.org/mission-and-vision). Similar to the analyses of research presented at PRCs (*Malički, Von Elm & Marušic, 2014*), we investigated the publication rate of research presented at ISMPP Annual Meetings.

## MATERIALS AND METHODS

This was a retrospective cohort study of ISMPP Annual Meeting abstracts (April 2009–April 2014).

Abstract metrics and data were obtained from *Current Medical Research and Opinion* (*CMRO*) Supplements (2009 onwards) and verified against ISMPP records. Submission and acceptance data were obtained from ISMPP. Corresponding full-text publications were identified by searching (17 December 2014) MEDLINE using the first, second, or last author surname and key terms from the title.

Abstracts were categorized based on author affiliations and study type. Publication rate was calculated as the percentage of presented abstracts published as full-text publications in peer-reviewed journals. Data were analyzed by Cochran Armitage trend test. Differences

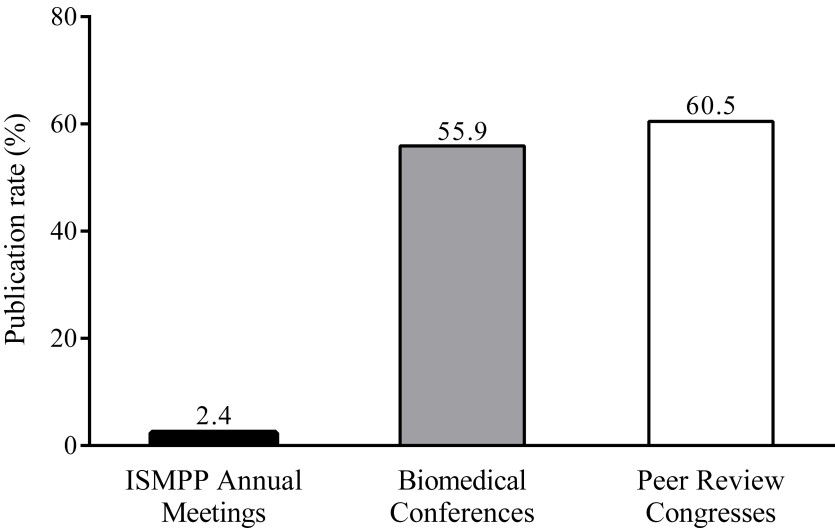

**Figure 1** Low publication rates from ISMPP Annual Meetings versus publication rates from biomedical conferences (*Scherer et al., 2015*) and Peer Review Congresses (*Maličsi, Von Elm & Marušić, 2014*). Abbreviation: ISMPP, International Society for Medical Publication Professionals.

in acceptance rate, abstracts published, study type, and contributor affiliations were considered significant at $P < .05$.

## RESULTS

Of 220 abstracts submitted to ISMPP Annual Meetings, 185 (84.1%) were accepted for presentation; of these, 164 (88.6%) were published in *CMRO*. Almost all of the research presented at ISMPP was never published in full in peer-reviewed journals; the publication rate was 2.4% (4/164; Fig. 1). Of the four research abstracts subsequently published as full papers, only 1 had been selected by ISMPP for an oral presentation (Table 1). The four publications were authored by 11 individuals who had advanced degrees (PhD or medical degree). Of these 11, eight were also Certified Medical Publication Professionals, signifying that they had passed an international exam on publication ethics and practices. All the authors worked as publication professional employees in medical communication or biopharmaceutical companies and, on average, had 20 peer-reviewed publications each.

Most abstracts were presented as posters (133/164; 81.1%). The remaining abstracts were, therefore, selected for oral presentations (i.e., 31/164; 18.9%). Abstracts described mainly observational (119/164; 72.6%) or opinion-based (37/164; 22.6%) research; interventional research was rare (6/164; 3.7%). Over time, the number of abstracts in *CMRO* increased significantly (15 in 2009 to 36 in 2014; $P = .02$); there were no changes in acceptance rate ($P = .44$) or study type (observational $P = .52$, interventional $P = .62$, opinion $P = .82$). Abstracts were submitted by researchers from the US (453/581; 78.0%), Europe (103/581; 17.7%), and the Asia-Pacific region (65/581; 11.2%). Most research was conducted by medical communication agencies (91/164; 55.5%), rather than healthcare companies (38/164; 23.2%).

**Table 1** The four research studies presented at ISMPP Annual Meetings that were published in peer-reviewed journals.

| ISMPP Annual Meeting | ISMPP decision on type of presentation | Research topic | MEDLINE-listed journal that published the research | Time from presentation to publication (months) |
|---|---|---|---|---|
| 2010 | Oral | Lack of involvement of medical writers and the pharmaceutical industry in publications retracted for misconduct: a systematic, controlled, retrospective study. | Current Medical Research and Opinion | 14 |
| 2011 | Poster | Publication misconduct and plagiarism retractions: a systematic, retrospective study. | Current Medical Research and Opinion | 18 |
| 2012 | Poster | Systematic review on the primary and secondary reporting of the prevalence of ghostwriting in the medical literature | BMJ Open | 27 |
| 2013 | Poster | Sponsor-imposed publication restrictions disclosed on ClinicalTrials.gov | Accountability in Research | 15 |

**Notes.**

Abbreviation: ISMPP, International Society for Medical Publication Professionals.

## DISCUSSION AND CONCLUSIONS

Research from ISMPP Annual Meetings has rarely been published in peer-reviewed journals. The publication rate (2.4%) is approximately 25-fold lower compared with research presented at biomedical conferences (55.9%) (*Scherer et al., 2015*) and PRCs (60.5%) (*Malički, Von Elm & Marušic, 2014*) (Fig. 1).

For publication professionals to join editors and the peer-review community in the quest to drive evidence-based improvements in publication practices (*Rennie & Flanagin, 2014*), they need to "practice what they preach" (i.e., design, conduct, and publish meaningful and robust research). Doing so would help the broader research community in its quest to improve publication practices and enable Good Publication Practice guidelines (*Battisti et al., 2015*), which many publication professionals follow (*Wager et al., 2014*), to be based on evidence, rather than expert opinion.

Our study has quantified, for the first time, the extent of nonpublication by publication professionals. Additional studies, ideally with qualitative research methods, will be required to investigate the reasons for the low publication rate. We speculate that lack of time may be one of the main drivers for nonpublication. A systematic review, which investigated the reasons why biomedical researchers don't publish, reported that lack of time was the most frequent, as well as the most important, reason provided for nonpublication (*Scherer et al., 2015*). In our experience, most publication professionals are not employed to design, conduct, and publish research on publication practices. Such research is likely to depend on considerable "after hours" time. Additional reasons may include lack of resources (e.g., limited funding to gain expert support from statisticians and professional medical writers), competing priorities (e.g., professional duties, personal responsibilities), and

uncertainty regarding the most important and feasible research projects to pursue. We also recognize that publication professionals may conduct the type of research (e.g., surveys and descriptive analyses on operational issues) that may not be appropriate nor robust enough for the peer-reviewed literature.

Our study has limitations. We focused only on ISMPP Annual Meetings. As our intent was to investigate how well publication professionals contribute to the evidence base affecting publication practices, we focused on the largest annual meeting for publication professionals. We recognize that publication professionals may attend other meetings, but these meetings may not be focused primarily on publication practices, may not have dedicated sessions for research-based presentations from publication professionals, and may not have been held as consistently or for as long as the ISMPP has held annual meetings. Our study is also limited by the focus on peer-reviewed publications included in the MEDLINE database (up until December 2014). We recognize that peer-reviewed publications may have appeared in other databases and after we conducted our search. We focused on MEDLINE because it was readily accessible and is one of the largest databases for medical research articles. We reasoned that those with an interest in medical research publication practices would search MEDLINE for relevant peer-reviewed articles.

In summary, the publication rate from research presented at ISMPP Annual Meetings is low. Publication professionals, who plan and prepare thousands of peer reviewed publications each year, should do more to contribute to evidence-based publication practices, including, and especially, their own.

## ACKNOWLEDGEMENTS

These findings were presented at the 11th Annual Meeting of the International Society for Medical Publication Professionals, 27–29 April 2015, Arlington, VA, USA.

### Funding

The independent statistical services were provided by Dr. Kathy Ruggiero (The University of Auckland, New Zealand), funded by Envision Pharma Group. The funders had no role in study design, data collection and analysis, decision to publish, or preparation of the manuscript.

### Grant Disclosures

The following grant information was disclosed by the authors:
Envision Pharma Group.

### Competing Interests

All authors are employees of Envision Pharma Group and members of not-for-profit associations supporting ethical publication practices. SS, CK, LW, and KW are Certified Medical Publication Professionals; KW serves on the ISMPP Board of Trustees.

## Author Contributions

- Luke C. Carey performed the experiments, wrote the paper, prepared figures and/or tables, reviewed drafts of the paper.
- Serina Stretton, Charlotte A. Kenreigh and Linda T. Wagner performed the experiments, wrote the paper, reviewed drafts of the paper.
- Karen L. Woolley conceived and designed the experiments, performed the experiments, wrote the paper, reviewed drafts of the paper.

## Data Availability

The raw data has been supplied as Data S1.

## Supplemental Information

Supplemental information for this article can be found online at http://dx.doi.org/10.7717/peerj.2011#supplemental-information.

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
