# Peer review of "High nonpublication rate from publication professionals hinders evidence-based publication practices"

_PeerJ, doi:10.7717/peerj.2011_

## Round 0.1 · original submission · Major Revisions

Please pay special attention to the comments of the Reviewer 1.

·

Basic reporting

Title - I would change the title in order to make it more specific:
High nonpublication rate from publication professionals hinders evidence-based publication - the International Society for Medical Publication Professionals experience

Introduction - I would add a sentence or two about the ISMPP and its mission and members to have the context.

Results - Figure 1 has to be explained in the text.
Results describing the characterisics of the posters should be in a table - Characteristics of posters presented at ISMPP annual meetings.

What about authors of the abstracts? where do they work? are they working as freelancers, for a company or for a journal? This should be stated and used for an explanations in the Discussion section. You could incorporate this in table 1.

The 4 published abstracts - what are they about? Are the corresponding authors active in publishing? Do hey have a publication record? (Please expalin in more detail about the pubslihed abstracts.) What is the publication record of the abstract authors? It would be ncie to know if they publish.

Discussion - This section is very condensed where I should expected more explanations about such a low publication index, even a speculation why publication professionals don't publish and some practical implications.
"should do more to contribute to evidence-based publication practices" - how?

Experimental design

Methods are well described.

Validity of the findings

The conclusions are appropriately stated, connected with the original question investigated, the limitations of the study are explained.

Additional comments

Dear authors, your manuscript is an interesting study about non publication rate from publication professionals. The Introuction is very well written and sound as are the Methods. However, the Results and the Discussion sections have to be written in more detail (you will see my recommendations). As a reader of this article I would like to know more about the authors of the abstracts (langiage professionals), and speculate about the possible reasons why they don't publish and conclude with some practical recomendations.

·

Basic reporting

The manuscript is clearly written.

Figure 1 does not add any useful information. I would omit.

As a reader unfamiliar with work presented at ISMPP conferences, I would appreciate some examples of the topics covered. These are available in the raw data but some examples in the main paper might be helpful: perhaps the four selected for oral presentation and/or the four that were published.

Two sentences in the Introduction seem superfluous: Conducting and publishing research on publication practices isn't easy, and the subsequent one.

In the introduction, the authors state 39% of presentations at peer review congresses have not been published, with no date range. In the figure and discussion, using the same reference, they give 60.5%.

In the results, it would be helpful to know the overall percentage of abstracts selected for oral presentation.

Experimental design

The research question was clearly stated and the investigation was well conducted.

Exact date of Medline search should be given.

Validity of the findings

The findings appear valid.

Additional comments

A well written paper.

·

Basic reporting

The study has the objective to raise awareness on the role of publication professionals and stimulate the production of peer reviewed articles about publication practices. The manuscript is well written, and the arguments clearly explained.
I would suggest to add some details on the International Society of Medical Publication Professionals (ISMPP) (aims, membership, etc) and how it relates to similar associations.
The limitation of the study is that research was performed only on one source: the abstracts of the annual meeting of the ISMPP and only Medline was searched to see if research described in the abstracts was then published in peer reviewed articles; in fact, also non medical peer reviewed journals might publish research on publication practices, such as Learned Publishing or the European Science Editing. The authors should explain why they searched only Medline and not other databases that may include peer reviewed articles on this subject.
The structure conforms to acceptable format. Supplementary material is useful.
The text is subdivided in coherent sections.
Better describe fig. 1 and improve its caption.

Experimental design

The research question is clearly defined and methods described with sufficient information to be reproducible.
Considering that the abstract lists refer to April 2009-November 2014 and that Medline was searched in December 2014 some abstracts may produce publication also after December 2014; this should be mentioned as well as the fact that other peer reviewed articles may have been published and not be included in Medline.

Validity of the findings

Data are correct; supplementary material is useful. Conclusions are connected with the original question and consistent with the title of the article.

---

## Round 0.2 · accepted · Accept

Thank you for successfully addressing all reviewers' comments.

·

Basic reporting

Abstract - maybe a little bit long Background, you could shorten it

Introduction - I would not start with Rennie and Flanagin... but it is upon the author . I like the Intro in the Abstract, it is more logical and nicer.

Results: "Of the 4 research abstracts subsequently published as full papers, only 1 had been selected by ISMPP for an oral presentation (Table 1). " maybe it would be better to formulate One out of four reserach abstracts was published.....

Experimental design

ok

Validity of the findings

ok

Additional comments

Dear author,

you have revised your manuscript according to my instructions and I am happy with the revisions. I have some comments that are minor, please look and decide whether you would like to make these changes, it is upon you.

·

Basic reporting

The authors have addressed all my points satisfactorily.

Experimental design

The authors have addressed all my points satisfactorily.

Validity of the findings

The authors have addressed all my points satisfactorily.

Additional comments

The authors have addressed all my points satisfactorily.

·

Basic reporting

The article has been revised according referees' suggestions. Referees' suggestions had many points in common; almost all of them were accepeted and if not, a reasonable explanation was provided.

Experimental design

No further comments

Validity of the findings

No further comments

Additional comments

The manuscript is now well written and consistent. Information is complete and fits to the journal topic.